# An Improved Image Compression Algorithm Using 2D DWT and PCA with Canonical Huffman Encoding

**DOI:** 10.3390/e25101382

**Published:** 2023-09-25

**Authors:** Rajiv Ranjan, Prabhat Kumar

**Affiliations:** 1Department of Information Technology, BIT Sindri, Dhanbad 828123, India; 2Department of Computer Science & Engineering, National Institute of Technology Patna, Patna 800005, India; prabhat@nitp.ac.in

**Keywords:** canonical Huffman coding (CHC), 2D discrete wavelet transform (2D DWT), hard thresholding, lossy image compression

## Abstract

Of late, image compression has become crucial due to the rising need for faster encoding and decoding. To achieve this objective, the present study proposes the use of canonical Huffman coding (CHC) as an entropy coder, which entails a lower decoding time compared to binary Huffman codes. For image compression, discrete wavelet transform (DWT) and CHC with principal component analysis (PCA) were combined. The lossy method was introduced by using PCA, followed by DWT and CHC to enhance compression efficiency. By using DWT and CHC instead of PCA alone, the reconstructed images have a better peak signal-to-noise ratio (PSNR). In this study, we also developed a hybrid compression model combining the advantages of DWT, CHC and PCA. With the increasing use of image data, better image compression techniques are necessary for the efficient use of storage space. The proposed technique achieved up to 60% compression while maintaining high visual quality. This method also outperformed the currently available techniques in terms of both PSNR (in dB) and bit-per-pixel (bpp) scores. This approach was tested on various color images, including Peppers 512 × 512 × 3 and Couple 256 × 256 × 3, showing improvements by 17 dB and 22 dB, respectively, while reducing the bpp by 0.56 and 0.10, respectively. For grayscale images as well, i.e., Lena 512 × 512 and Boat 256 × 256, the proposed method showed improvements by 5 dB and 8 dB, respectively, with a decrease of 0.02 bpp in both cases.

## 1. Introduction

With the phenomenal rise in the use of digital images in the Internet era, researchers are concentrating on image-processing applications [1,2]. The need for image compression has been growing due to the pressing need to minimize data size for transmission. This has become particularly necessary due to the restricted capacity of the Internet. The primary objectives of image compression are to store large amounts of data in a small memory space and to transfer data quickly [2].

There are primarily two types of image compression methods: lossless and lossy. In lossless compression, the original and the reconstructed images remain exactly the same. On the other hand, in lossy compression, notwithstanding its extensive application in many domains, there can be data loss to a certain extent for greater reduction of redundancy. In lossy compression, the original image is first forward transformed before the final image is quantized. The compressed image is then produced using entropy encoding. This process is shown in Figure 1.

Lossy compression can additionally be classified into two primary methods [3,4]:

Firstly, there are direct image compression methods, which are applied for sampling an image in a spatial domain. These methods comprise techniques such as block truncation (block truncation coding (BTC) [5], absolute moment block truncation (AMBTC) [6], modified block truncation coding (MBTC) [7], improved block truncation coding using K-means quad clustering (IBTC-KQ) [8], adaptive block truncation coding using edge-based quantization approach (ABTC-EQ) [9]) and vector quantization [10].

Secondly, there are image transformation methods, which include singular value decomposition (SVD) [11], principal component analysis (PCA) [12], discrete cosine transform (DCT) [13] and discrete wavelet transform (DWT) [14]. Through these methods, image samples are transformed from the spatial domain to the frequency domain, thereby concentrating the energy of the image in a small number of coefficients.

Presently, researchers are noticeably turning to the DWT transformation tool due to its pyramidal or dyadic wavelet decomposition properties [15]. It enables high compression and helps produce superior-quality reconstructed images. The present study demonstrates the benefits of the DWT-based strategy using canonical Huffman coding. In their preliminary work, the present authors explained this aspect of the entropy encoder [16]. In course of the analysis, a comparison of canonical Huffman coding with basic Huffman coding showed that the former has a smaller code-book size and accordingly requires less processing time.

In the present study, for the standard test images, the issue of enhancing the compression ratio was addressed by improving quality of the reconstructed image and by thoroughly analyzing the necessary parameters, such as PSNR, SSIM, CR and BPP. PCA, DWT, normalization, thresholding and canonical Huffman coding methods were employed to achieve high compression with excellent image quality. During the present study, canonical Huffman coding proved to be superior to both Huffman and arithmetic coding, as explained in Section 3.4.

The present authors developed a lossy compression technique during the study using the PCA [12], which proved to be marginally superior to the SVD method [17] and DWT [16] algorithms for both grayscale and color images. Canonical Huffman coding [16] was used to compress the reconstructed image to a great extent. The authors also compared the parameters obtained in their proposed method with those provided in the block truncation [18] and the DCT-based approaches [19].

In this study, PCA-DWT-CHC extends our previously reported work (DWT) [16]. Our proposed method uses the Haar wavelet transform to decompose images to a single level and then incorporates PCA with DWT to improve performance. In the previous work, images were decomposed up to three levels using a Haar wavelet, and PCA was not included as a pre-processing compression method. Here, the proposed method yields high-quality images with a high compression ratio and requires less computing time (by an average of 45%) than our previously reported work. In the process of the study, the authors examined several frequently cited images in the available literature. Slice resolutions of 512 × 512 and 256 × 256 were used, which are considered to be the minimum standards in the industry [20]. The present authors also calculated the compression ratio and the PSNR values of their methods and compared them to the other research findings [3,5,6,7,8,9,16,20].

This study was based on the following structure: After the introduction in Section 1, a review of the literature is presented in Section 2. Section 3 discusses the approach adopted in the present study and also analyzes a number of critical concepts. Section 4 details the proposed algorithm. The parameters for performance evaluation are explained in Section 5. Section 6 presents the experiment findings, while Section 7 marks the conclusion.

## 2. Literature Review

An overview of several published works on this subject highlights various other methods that have so far been presented by many other researchers. One approach that has gained considerable attention in recent years among the research communities is a hybrid algorithm that combines DWT with other transformation tools [10]. S M Ahmed et al. [21] explained in detail their method of compressing ECG signals using a combination of SVD and DWT. Jayamol M. et al. [8] presented an improved method for the block truncation coding of grayscale images known as IBTC-KQ. This technique uses K-means quad clustering to achieve better results. Aldzjia et al. [22] introduced a method for compressing color images using the DWT and genetic algorithms (GAs). Messaoudi et al. [3] proposed a technique called DCT-DLUT that involves using the discrete cosine transform and a lookup table known as DLUT to demarcate the difference between the indices. It is a quick and effective way to compress lossy color images. 

Paul et al. [10] proposed a technique, namely DWT-VQ (discrete wavelet transform-Vector Quantization), for generating an YCbCr image from an RGB image. This technique compresses images while maintaining their perceptual quality in a clinical setting. A K Pandey et al. [23] presented a compression technique that uses the Haar wavelet transform to compress medical images. A method for compressing images using the discrete Laguerre wavelet transform (DLWT) was introduced by J A Eleiwy [24]. However, this method concentrates only on approximate coefficients from four sub-bands of the DLWT post-decomposition. As a result, this approach may affect the quality of the reconstructed images. In other words, maintaining a good image quality while achieving a high compression rate can prove to be considerably challenging in image compression. Moreover, J. A. Eleiwy did not apply the peak signal-to-noise ratio (PSNR) or the structural similarity index measure (SSIM) index to evaluate the quality of the reconstructed image.

M. Alosta et al. [25] examined the arithmetic coding for data compression. They measured the compression ratio and the bit rate to determine the extent of the image compression. However, their study did not assess the quality of the compressed images, specifically the PSNR or SSIM values, which correspond to the compression rate (CR) or bits per pixel (BPP) values.

R. Boujelbene et al. [20] have shown that the NE-EZW algorithm provides a triple tradeoff between the number of symbols, image size, and reconstructed image quality. It builds upon the EZW coding approach and outperforms both the JPEG 2000 [26] and SPIHT [27] compression algorithms.

S. Singh et al. [28] validated SPIHT’s superiority over JPEG [29] in medical image datasets for image quality at the same bit rate.

## 3. Fundamental Concepts

Various phases of the suggested method for the present study are outlined in this section. These include canonical Huffman coding, DWT and PCA. Transformation is a mathematical process through which a function, considered to be an input, is mapped. Transformation can extract hidden or valuable data from the original image. Moreover, in comparison with the original data, the transformed data may be more amenable to mathematical operations. Therefore, transformation tools are a significant means for image compression.

The most widely used transformation methods include the Karhunen–Loeve transform (KLT) [30], Walsh–Hadamard transforms (WHTs) [31], SVD [11], PCA [12], DCT [13], DWT [14] and integer wavelet transform (IWT) [32].

The DCT method is commonly used for compressing images. However, it may result in image artifacts when compressed with JPEG. Moreover, DCT does not have the multi-resolution transform property. In all these respects, DWT is superior [33]. With DWT, one can obtain the resulting filtered image after going through various levels of discrete wavelet decomposition. One can also gather statistics from the frequency domain for the following procedure via multi-level wavelet decomposition. Following the compression, by combining noise reduction and information augmentation, better image reconstruction can be ensured [2].

Hence, DWT was the preferred method for image compression during the present study [14]. Because of its high energy compaction property and lossy nature, this technique can remove unnecessary data from an image to achieve the desired compression level for images. It produces wavelet coefficients iteratively by dividing an image into low-pass and high-pass components. These wavelet coefficients de-correlate pixels while the canonical Huffman coding eliminates redundant data.

### 3.1. Principal Component

Principal components are a small number of uncorrelated variables derived from several correlated variables by means of the PCA [12] transformation technique. The PCA technique determines the finer points in the data to highlight their similarities and differences. Once the patterns are established, datasets can be compressed by reducing their dimensions without losing the basic information. Therefore, the PCA technique is suitable for image compression with minimal data loss.

The idea of the PCA technique is to take only the values of the principal components and use them to generate other components.

In short:❖PCA is a standard method for reducing the number of dimensions.❖The variables are transformed into a fresh set of data, known as primary components. These principal components are combinations of initial variables in linear form and they are orthogonal.❖The first principal component accounts for the majority of the potential variation in the original data.❖The second principal component addresses the data variance.

#### 3.1.1. Mathematical Concepts of PCA

The PCA algorithm: The following steps make the PCA Algorithm:
Step-01:Obtaining data.Step-02:Determining the mean vector (µ).Step-03:Subtracting the mean value from the data.Step-04:Performing a covariance matrix calculation.Step-05:Determining the eigenvalues and eigenvectors of the covariance matrix.Step-06:Assembling elements to create a feature vector.Step-07:Creating a novel data set.

#### 3.1.2. Mathematical Example

Two-dimensional patterns have to be taken into account, i.e., (2, 1), (3, 5), (4, 3), (5, 6), (6, 7), and (7, 8). This must be followed by principal component calculation.


*Step-01:*


Data are obtained. x1 = (2, 1), x2 = (3, 5), x3 = (4, 3), x4 = (5, 6), x5 = (6, 7) & x6 = (7, 8).

The vectors provided are - $\left[\begin{array}{c} 2\\ 1 \end{array}\right]$ $\left[\begin{array}{c} 3\\ 5 \end{array}\right]$ $\left[\begin{array}{c} 4\\ 3 \end{array}\right]$ $\left[\begin{array}{c} 5\\ 6 \end{array}\right]$ $\left[\begin{array}{c} 6\\ 7 \end{array}\right]$ $\left[\begin{array}{c} 7\\ 8 \end{array}\right]$


*Step-02:*


The mean vector (µ) is identified.
Mean vector (µ) = ((2 + 3 + 4 + 5 + 6 + 7)/6, (1 + 5 + 3 + 6 + 7 + 8)/6) = (4.5, 5)
$$Mean\;vector\left(\mu \right)=\left[\begin{array}{c} 4.5\\ 5 \end{array}\right]$$


*Step-03:*


The mean vector (µ) is subtracted from the data.
x1 − µ = (2 − 4.5, 1 − 5) = (−2.5, −4)

Similarly, other feature vectors are obtained.

After removing the mean vector (µ), the following feature vectors (x_i_) are obtained:

$\left[\begin{array}{c} -2.5\\ -4 \end{array}\right]$ $\left[\begin{array}{c} -1.5\\ 0 \end{array}\right]$ $\left[\begin{array}{c} -0.5\\ -2 \end{array}\right]$ $\left[\begin{array}{c} 0.5\\ 1 \end{array}\right]$ $\left[\begin{array}{c} 1.5\\ 2 \end{array}\right]$ $\left[\begin{array}{c} 2.5\\ 3 \end{array}\right]$


*Step-04:*


A covariance matrix calculation is performed.

The covariance matrices are provided by-
$$Cov\;Mat=\frac{\sum \left(x_{i}-\mu \right){\left(x_{i}-\mu \right)}^{t}}{n}$$

Now, $m_{1}=\left(x_{1}-\mu \right){\left(x_{1}-\mu \right)}^{t}=\left[\begin{array}{c} -2.5\\ -4 \end{array}\right]\left[\begin{array}{cc} -2.5 & -4 \end{array}\right]=\left[\begin{array}{cc} 6.25 & 10\\ 10 & 16 \end{array}\right]$

Similarly, the value of $m_{2}\ldots m_{6}$ is calculated.

The covariance matrix is now equal to (m1 + m2 + m3 + m4 + m5 + m6)/6. 

The matrices above are added and divided by 6: $$CovMat=\left[\begin{array}{cc} 2.92 & 3.67\\ 3.67 & 5.67 \end{array}\right]$$


*Step-05:*


The eigenvalues and eigenvectors of the covariance matrix are determined.

A value is considered to be an eigenvalue (*λ*) for a matrix M if it solves the defining equation |M − *λ*| = 0.

Hence, one obtains:$\left|\begin{array}{cc} 2.92-\lambda & 3.67\\ 3.67 & 5.67-\lambda \end{array}\right|=0$, 

By resolving this quadratic problem = 8.22, 0.38 is obtained.

Hence, eigenvalues *λ*1 and *λ*2 are 8.22 and 0.38, respectively. 

It is obvious that the second eigenvalue is much smaller than the first eigenvalue. 

Hence, it is possible to exclude the second eigenvector. The primary component is the eigenvector that corresponds to the highest eigenvalue in the given data set. As a result, the eigenvector is located matching eigenvalue *λ*1. The eigenvector is determined by applying the following equation:MX=λX

*X* = Eigenvector, *M* = Covariance Matrix and *λ* = Eigenvalue.

By changing the values in the aforementioned equation, X2 = 1 and X1 = 0.69 are obtained. Then, these numbers are divided by the square root of the sum of their squares. The eigenvector V is
$$\left[\begin{array}{c} x_{1}\\ x_{2} \end{array}\right]=\left[\begin{array}{c} 0.566\\ 0.821 \end{array}\right]$$

Hence, the principal component of the presented data set is
$$\left[\begin{array}{c} x_{1}\\ x_{2} \end{array}\right]=\left[\begin{array}{c} 0.566\\ 0.821 \end{array}\right]$$

### 3.2. Discrete Wavelet Transform

#### The Operational Principle of DWT

The data matrix of the image is split into four sub-bands, i.e., LL (low-pass vertical and horizontal filter), LH (low-pass vertical and high-pass horizontal filter), HL (high-pass vertical and low-pass horizontal filter) and HH (high-pass vertical and horizontal filter). These sub-bands are used to apply the wavelet transform in computing (DWT [14] and Wavelet [HAAR] [19]). The logic behind the decomposition of the image into the four sub-bands is explained in Figure 2.

The process involves dividing the image into rows and columns after convolution. The wavelet decomposition and reconstruction phases make the DWT. The image input undergoes a process of convolution that includes both low- and high-pass reconstruction phases. Figure 3a describes a one-level DWT decomposition. In Figure 3b, the up arrow denotes the up-sampling procedure. The wavelet reconstruction is the opposite of the wavelet decomposition.

For the data processing, various wavelet families are commonly used such as Haar (“haar”), Daubechies (“db”), Coiflets (“coif”), Symlets (“sym”), Biorthogonal (“bior”) and Meyer (“meyer”) [23]. During the present study, the Haar wavelet transform was applied due to its comparatively modest computational needs [19].

### 3.3. Thresholding

#### Hard Thresholding

The hard-thresholding method is used frequently in image compression. The hard-threshold function works by $\varphi_{T}\left(x\right)=x\cdot 1\left\{\left|x\right|>T\right\}$ keeping the input value if it is greater than the set threshold T. If the input value is less than or equal to the threshold, it is set at zero [34].

### 3.4. Entropy Encoder 

#### Canonical Huffman Coding

Canonical Huffman coding [16,35] is a significant subset of regular Huffman coding and has several advantages over other coding schemes (Huffman, arithmetic). Its advantages include faster computation times, superior compression and higher reconstruction quality. Many researchers prefer working with this coding because of these advantages. The information required for decoding is compactly stored since the codes are in lexicographic order.

For instance, if the Huffman code for five bits is “00010,” only five will be used for canonical Huffman coding, equaling the entire number of bits available in the Huffman code [36].

## 4. Proposed Method 

In the course of the present study, various approaches for compressing images were examined, including that of transforming RGB color images into $YC_{b}C_{r}$ color images [37], PCA transformation, wavelet transformation and extra processing by using thresholding, normalization and canonical Huffman coding.

### 4.1. Basic Procedure 

During the present study, in order to compress the image, the PCA approach was applied first. Next, the output of PCA was decomposed using DWT. Finally, the image was further decomposed using canonical Huffman encoding. In order to break down 8-bit/24-bit key images with 256 × 256 and 512 × 512 pixel sizes, a one-level Haar wavelet transform was used. 

### 4.2. Pca Based Compression

The PCA procedure involves mapping from an n-dimensional space to a k-dimensional space by applying orthogonal transformations (k < n). The principal components, which are unique orthogonal features in this case, are the k-dimensional features that include most of the characteristics of the original data set. Because of this advantage, it is used in image compression.

PCA is a reliable image compression technique that ensures nominal information loss. In comparison with the SVD approach, the PCA method produces better results [17]. The Algorithm 1 [12] based on PCA is shown in below.
**Algorithm** **1**: PCA_Algorithm [12]*Encoding**Input*: The image Fx,y, Fx,y=f0,0⋯fm−1⋮⋱⋮fn−1,0⋯fn−1,m−1Here, the values *x* and *y* represent the coordinates of individual pixels in an image. Depending on the type, the value fx,y corresponds to the color or gray level.*Step 1*: Image normalization has to be performed.The normalization is carried out on the image data set
Fx,y.Fnormalizedx,y=f0,0⋯fm−1⋮⋱⋮fn−1,0⋯fn−1,m−1−f¯0,0…f¯0,m−1Here, f¯0,0…f¯0,m−1 is the column vector containing the mean value for y1 to ym.*Step 2*: Computation of covariance matrix of Fnormalizedx,y is performed.covx,y=Fnormalizedx,y × Fnormalizedx,yTm−1Here, *m* is the number of element y.*Step 3*: Computation of Eigenvectors and Eigenvalues of covx,y is performed.Using the SVD equation AT=covx,y=UD2UT, the eigenvectors and eigenvalues are calculated.Here, “U” represents the eigenvectors of “AAT”, while the squared singular values in “D” are the eigenvalues of “AAT”. The eigenvector matrix denotes the principal feature of image data, i.e., the principal component.*Output*: Image data with reduced dimension:Ftransformedx,y=UTFnormalizedx,yHere, UT is the transpose of the eigenvectors matrix and Fnormalizedx,y is the adjusted original image datasets.It can also be expressed as:Ym×k=Un×kTXm×nHere, “*m*” and “*n*” represent in the matrix, while “*k*” represents the number of principal components with k<m,n.*Decoding*By reconstructing the image data, one obtainsX^m×n=Un×kYm×kIn PCA, the compression ratio (ρ) [12] is calculated as:ρ=n×nm×k+n×k+n

### 4.3. Dwt-Chc Based Compression

The DWT details show zero mean and a slight variation. The more significant DWT coefficients are used and the less significant ones are discarded by using canonical Huffman coding. The Algorithm 2 based on DWT is presented below.
**Algorithm** **2:** DWT_CHC Algorithm [16]*Input*: An image in grayscale GA,B of size A×B
*Output*: A reconstruction of a grayscale image RA,B of size A×B*Encoding of Image*
*Step 1*:The DWT is applied to separate the grayscale image GA,B into lower and higher sub-bands.*Step 2*:The equation an = ad−aminamax−amin, is applied to normalize the lower and upper sub-bands in the range of (0, 1), where a is the coefficient matrix of the image GA,B, ad is the data to be normalized, and amax nd amin are the maximum and minimum intensity values, respectively. *Step 3*:Hard thresholding on the higher sub-band is used to save the important bits and discard the unimportant ones.*Step 4*:To acquire the lower and higher sub-band coefficients, the lower sub-band coefficient is assigned to the range of 0 to 127 and the higher sub-band coefficient is assigned to the range of 0 to 63.*Step 5*:Canonical Huffman coding is applied to each band.*Step 6*:The compressed bit streams are obtained.

*Decoding of Image*

*Step 1*:The compressed bit streams are taken as an input.*Step 2*:The reverse canonical Huffman coding process is applied to retrieve the reconstructed lower and higher sub-band coefficients from the compressed bit streams of the approximate and detail coefficients.*Step 3*:To obtain the normalized coefficients for the lower and higher sub-bands, their respective coefficients are divided by 127 and 63. *Step 4*:The following equation is applied to perform inverse normalization on the normalized lower and higher sub-bands.
$a_{d}$ = $a_{n}$ × ($a_{\max}$−$a_{\min}$) + $a_{\min}$
*Step 5*:Inverse DWT is applied to obtain the rebuilt image $R\left(A,B\right)$.

### 4.4. Pca-Dwt-Chc-Based Image Compression 

This method first involves compression of the image through the PCA, followed by decomposing the gray scale/color image by using a one-level Haar wavelet transform. One achieves approximate and detailed images by applying this method. To produce a digital data sequence, the approximation coefficients have to be normalized and encoded with canonical Huffman coding. Moreover, during normalization of the detail coefficients, any insignificant coefficients are removed through hard thresholding. Finally, binary data are obtained by using canonical Huffman coding.

The final compressed bit stream is created by combining all the binary data. This stream is then divided into approximate and detailed coefficient binary data to reconstruct the image. The qualitative loss becomes apparent only after a certain point by eliminating certain principal components. This entire procedure is termed the DWT-CHC method. During the present study, the proposed strategy was found to work better when the PCA-based compression technique was used with DWT-CHC as a part of the lossy method. The DWT outperformed PCA in terms of compression ratios while the PCA outperformed the DWT in terms of the PSNR values. An evaluation of the necessary number of bits yielded the CR value for the PCA algorithm. 

During the present experimentation, initially, the image was compressed by using the PCA. The approximate image was further compressed by using DWT-CHC. Accordingly, the image was initially decomposed using PCA, then a few principal components were removed. The reconstructed image was then computed. Next, the reconstructed image was used as the input image for the DWT-CHC segment of the proposed method. 

When several primary components were dropped from the PCA segment of the proposed method, the compression ratio was found to be higher. The overall CR value was obtained by multiplying the CR values of the PCA and the DWT-CHC. 

To analyze an image, it is first decomposed, applying a Haar wavelet to its approximation, horizontal, vertical and diagonal detail coefficients. Next, the approximation and the detail coefficients are coded with DWT-CHC. Encoding refers to the compression process and decoding refers to the simple process of reversing the encoding stages from which the reconstructed image is derived. After quantization, the image is rebuilt using the inverse DWT-CHC of the quantized block. 

This approach combines the PCA and the DWT-CHC to reach its full potential. It uses PCA, DWT and canonical Huffman coding to achieve a high compression ratio while maintaining excellent image quality. The structural layout of the proposed image compression approach is shown in Figure 4a–e.

The steps in the suggested method are as follows:


*Encoding:*


*Step 1*: (i) For a *C(x*, *y*) grayscale image with an *x* × *y* pixel size to be derived by using the PCA method, as the first step, the image is decomposed first to obtain the principal component. 

(ii) If the image is in color, the color transform is used to change the RGB data into $YC_{r}C_{b}$ using the formula. $\left[\begin{array}{c} Y\\ C_{r}\\ C_{b} \end{array}\right]=\left[\begin{array}{ccc} 0.299 & 0.587 & 0.114\\ -0.169 & -0.331 & 0.500\\ 0.500 & -0.419 & -0.081 \end{array}\right]\left[\begin{array}{c} R\\ G\\ B \end{array}\right]$

To demarcate the principal component from the $YC_{r}C_{b}$ image, PCA decomposition is carried out.

*Step 2*: The image is reconstructed by utilizing these principal components. Accordingly, for compression, only the principal components are considered.

*Step 3*: The compression ratio is obtained.

*Step 4*: The decomposition level is set at 1.

*Step 5*: By utilizing the HAAR wavelet, the DWT generates four output matrices: LL (the approximate coefficients) and LH, HL and HH (the detail coefficients). These matrices consist of three components: vertical, horizontal and diagonal details. 

*Step 6*: To obtain bit streams, the DWT-CHC algorithm is applied to these coefficients (compressed image).

*Step 7*: The compression ratio is calculated.

*Step 8*: To determine the final compression ratio of an image, the outputs from Steps 3 and 7 are multiplied.


*Decoding:*


*Step 1*: The DWT-CHC approach is applied in reverse to obtain the approximate and detailed coefficients.

*Step 2:* A reconstructed image is created.

*Step 3:* The PSNR value is determined. 

The following flowchart further explains this idea:

## 5. Performance Assessment 

A few of the parameters listed below can be used to gauge the efficacy of the lossy compression strategy.

Compression ratio (CR): CR [38] is a parameter that measures compressibility.

Mathematically, $CR=\frac{S_{original}}{S_{compressed}}$

where $S_{original}$ = the size of the original image data, $S_{compressed}$ = a measure for the size of the compressed image data (in bits).

Bitrate (BPP): the BPP equals 24/CR for color images and 8/CR for grayscale images.

Peak signal-to-noise ratio (PSNR): This is a common metric for assessing the quality of the compressed image. Typically, the PSNR for 8-bit images is formulated as: [39]
(1)$$PSNR\left(dB\right)={10\log}_{10}\left(\frac{255^{2}}{MSE}\right)$$
where 255 is the highest value that the image signal is capable of achieving. The term “*MSE*” in Equation (1) refers to the mean squared error of the image, written as
$$MSE=\frac{1}{m}\sum_{x}\sum_{y}{\left(f\left(x,y\right)-F\left(x,y\right)\right)}^{2}$$

Here, the variable “*m*” represents the total number of pixels in the image. *F* (*x*, *y*) refers to the value of each pixel in the compressed image, while *f* (*x*, *y*) represents the value of each pixel in the original image.

Structural similarity index (SSIM): This is a process for determining how similar two images can be [40].
$$\mathrm{Luminance}\;\mathrm{change},\;l\left(x,y\right)=\frac{2\mu_{x}\mu_{y}+c_{1}}{\mu_{X}^{2}+\mu_{X}^{2}+c_{1}}$$
$$\mathrm{Contrast}\;\mathrm{change},\;c\left(x,y\right)=\frac{2\sigma_{x}\sigma_{y}+c_{2}}{\sigma_{X}^{2}+\sigma_{X}^{2}+c_{2}}$$
$$\mathrm{Structural}\;\mathrm{change},s\left(x,y\right)=\frac{\sigma_{xy}+C_{3}}{\sigma_{x}\sigma_{y}+C_{3}}$$

Here, SSIM can be evaluated as:$$SSIM\left(x,y\right)=l\left(x,y\right)\cdot c\left(x,y\right)\cdot s\left(x,y\right)$$


*y* represents the image that was recreated and *x* represents the original image. 

$\mu_{x}$ = average of x, $\mu_{y}$ = average of y

$\sigma_{x}$ = variance of x, $\sigma_{y}$ = variance of y

Two variables, $c_{1}$ and $c_{2}$, are used to stabilize a division with a weak denominator.
$$c_{1}={\left(k_{1}L\right)}^{2},\;c_{2}={\left(k_{2}L\right)}^{2},\;c_{3}=\frac{c_{2}}{2}$$

$k_{1}=0.001$, $k_{2}=0.002$ as a rule.

In this case, the pixel values range from 0 to 255 and are represented by L. The SSIM index is generated as a consequence, ranging from −1 to 1.

## 6. Experiment Result

The outcomes of the experiment for image compression, utilizing the PCA-DWT-CHC hybrid approach, are presented in this section. Additionally, a comparison between the suggested approach and other available methods, such as (BTC [5], AMBTC [6], MBTC [7], IBTC-KQ [8], ABTC-EQ [9], DWT [16], DCT-DLUT [3] and NE-EZW [20]) is made. All experiments in this connection were conducted using the 512 × 512 and 256 × 256 input images (8-bit grayscale images, which are Lena, Barbara, Baboon, Goldhill, Peppers, Cameraman, Boat and 24-bit color images, i.e., Airplane, Peppers, Lena, Couple, House, Zelda and Mandrill). The images are presented in Figure 5 and Figure 6.

All experiments were run in the interim on the MATLAB software (Version 2013a) platform using the hardware configurations of an Intel Core i3-4005U processor. It had 1.70 GHz, 4.00 GB of RAM and Windows 8.1 Pro 64-bit as the operating system.

The compression performance of images for various approaches is shown in the next part, which is based on visual quality evaluation and objective image quality indexes, i.e., PSNR, SSIM, CR and BPP.

Two parameters, namely CR and BPP, reflect certain common aspects of image compression. The PSNR and SSIM are used to assess the quality of the compressed image. Greater PSNR and SSIM values indicate better image reconstruction, whereas higher compression ratios and lower bitrates indicate enhanced image compression.

For this study, the predictive approach was used to determine the threshold values, which were TH = 0.10. For both color (256 × 256 × 3 and 512 × 512 × 3) and grayscale (256 × 256 and 512 × 512) images, the principal component values of 25, 25, 200 and 400 were taken, respectively, to reconstruct the image.

### 6.1. Visual Performance Evaluation of Proposed Pca-Dwt-Chc Method

Based on the quality of the reconstructed images, the proposed hybrid PCA-DWT-CHC image compression method was compared to the other methods that are available presently. Figure 7b,d and Figure 8b,d present the reconstructed images for comparison of the visual quality on the basis of the PSNR values, i.e., 34.78 dB, 33.31 dB, 33.43 dB and 37.99 dB with CR = 4.41, 4.04, 5.15 and 4.45 for the input grayscale images. The images are respectively titled as “lena.bmp” and “barbara.bmp” (size 512 × 512) and “cameraman.bmp” and “boat.bmp” (size 256 × 256). Again, Figure 9a,b and Figure 10a,b display the reconstructed image for a comparison of visual quality on the basis of the PSNR values, i.e., 47.57 dB, 47.99 dB, 54.60 dB and 53.47 dB, with compression factors (in BPP) of 0.27, 0.32, 0.69 and 0.70, respectively, for the input color images. The respective titles of the images are “airplane.bmp,” and “peppers.bmp” (size 512 × 512 × 3) along with “couple.bmp” and “house.bmp” (size 256 × 256 × 3).

Figure 7, Figure 8, Figure 9 and Figure 10 demonstrate that the proposed hybrid PCA-DWT-CHC method yielded the reconstruction of a superior-quality image as compared to other image compression methods with regard to all the input images. Based on the visual quality in various standard test images, one could conclude that the proposed hybrid PCA-DWT-CHC method is more efficient in reconstructing images compared to the other available methods.

### 6.2. Objective Performance Evaluation of Proposed Pca-Dwt-Chc Method

#### 6.2.1. Tabular Results for Comparative Analysis of Proposed Method

According to the experiment results, the suggested PCA method, followed by the DWT-CHC method, proved to be better in terms of the PSNR, SSIM, BPP and CR values when compared to the other methods, as shown in Table 1, Table 2 and Table 3. In other words, from Table 1, Table 2 and Table 3, the proposed method is superior to the BTC [5], AMBTC [6], MBTC [7], IBTC-KQ [8], ABTC-EQ [9], DWT [16], DCT-DLUT [3] and NE-EZW [20] processes for working on grayscale and color images. This is established by the fact that among all the different compression methods, the PSNR and SSIM values in the proposed method were found to be the highest. The CR value was also the highest in the proposed method compared to the other methods. On the other hand, with regard to the color images, the bitrate values in the proposed method turned out to be the lowest when compared to the other available methods.

#### 6.2.2. Graphical Representation for Comparative Analysis of Proposed Method

The graphs in Figure 11, Figure 12, Figure 13, Figure 14, Figure 15, Figure 16, Figure 17 and Figure 18 show the PSNR, SSIM, CR and compression factor (in bpp) results for the eight grayscale images. Figure 19, Figure 20, Figure 21 and Figure 22, on the other hand, present graphs for the PSNR and compression factor (in bpp) for the eight other color images. After comparing the data from these graphs with the data of various other available techniques, the former proved to be more effective.

Figure 11, Figure 15, Figure 19 and Figure 21 display the PSNR characteristics, while Figure 12 and Figure 16 show the SSIM index. The CR values are found in Figure 13 and Figure 17 and the compression factor (in bpp) can be seen in Figure 14, Figure 18, Figure 20 and Figure 22. It is evident from the four PSNR plots that the proposed hybrid PCA-DWT-CHC method performs better than the DWT and other existing approaches in terms of the PSNR values. 

The proposed method is successful in enhancing image compression without compromising image quality. In comparison to the other methods, such as NE-EZW, DCT-DLUT, DWT, ABTC-EQ, IBTC-KQ, MBTC, AMBTC and BTC, this process was observed to maintain or even improve the original image quality. 

In other words, the suggested hybrid method established with data that it ensures superior image reconstruction compared to the other methods. It has also shown improvement in image compression, as indicated by its higher CR values and lower compression factor (in bpp). Figure 16 demonstrates that the proposed hybrid PCA-DWT-CHC method ensures the highest SSIM values for all the test images. In other words, it is able to reconstruct all the images with greater similarity to the original ones compared to the other available methods.

### 6.3. Time Complexity Analysis of Proposed Pca-Dwt-Chc Method 

The speed of the proposed method’s encoding and decoding process is essential for real-time compression. Its time complexity needs to be explained to give a clear idea of whether the proposed method can be used in real-time applications. During the present study, the total time required for the proposed approach to encode and decode data was assessed to ensure proper evaluation of its time complexity. The average time requirements were calculated and compared for analysis purposes. Table 4 presents the average time requirements of the proposed method and DWT [16] for encoding and decoding processes. According to Table 4, the proposed method coder is quicker than DWT method coders. Compared to the DWT method, encoding and decoding the four test images took 76.5826, 22.5475, 24.5913 and 50.8211 less time. The proposed hybrid PCA-DWT-CHC transform proves to be faster than the DWT method. Therefore, one could claim the proposed method to be effective for applications that require real-time image compression.

## 7. Conclusions

The objective of the present study was to develop a method for superior image quality and compression. By combining PCA, DWT and canonical Huffman coding, a new approach was developed for compressing images. Accordingly, the proposed method was able to outperform the existing methods, such as BTC, AMBTC, MBTC, IBTC-KQ, ABTC-EQ, DWT, DCT-DLUT and NE-EZW. Lower bit rates and better PSNR, CR and SSIM values indicate improved image quality. A comparison of the PSNR, SSIM, CR and BPP values resulting from the proposed technique with those of the other available approaches confirmed the former’s superiority.

The findings from the objective and subjective tests prove that the newly developed approach offers a more efficient image compression technique compared to the existing approaches. For example, when working with the grayscale images of 256 × 256 and 512 × 512 resolutions, we secured improved results in metrics with regard to the PSNR, SSIM, BPP and CR. Again, in the case of the color images of 256 × 256 × 3 and 512 × 512 × 3 resolutions, improved PSNR and lower BPP results were noted. Therefore, one could conclude that the present research has the potential to greatly improve the storage and transmission quality of image data across digital networks.

## Figures and Tables

**Figure 1 entropy-25-01382-f001:**
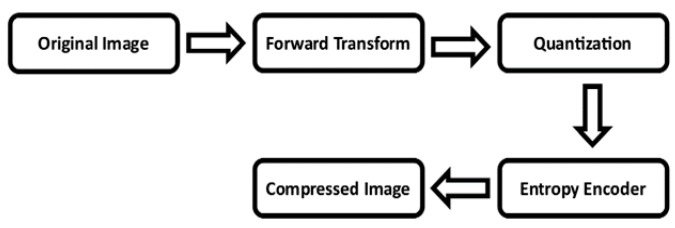
Lossy Image Compression Block Diagram in General.

**Figure 2 entropy-25-01382-f002:**
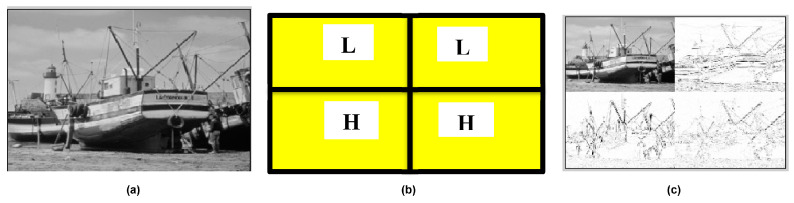
Decomposition of discrete wavelet transform: (**a**) input image, (**b**) image sub-bands, and (**c**) 1-Level DWT decomposition.

**Figure 3 entropy-25-01382-f003:**
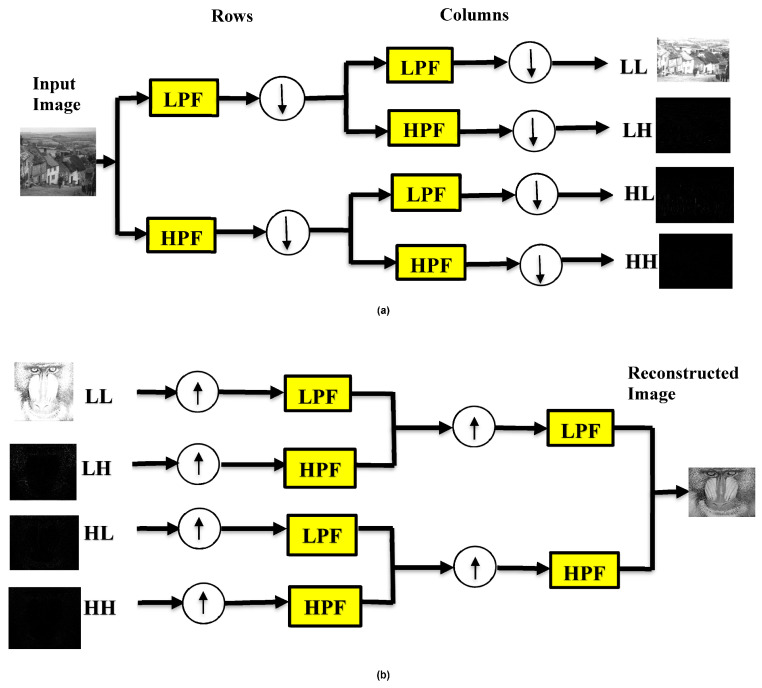
(**a**) Decomposition and (**b**) reconstruction of a one-level discrete wavelet transform.

**Figure 4 entropy-25-01382-f004:**
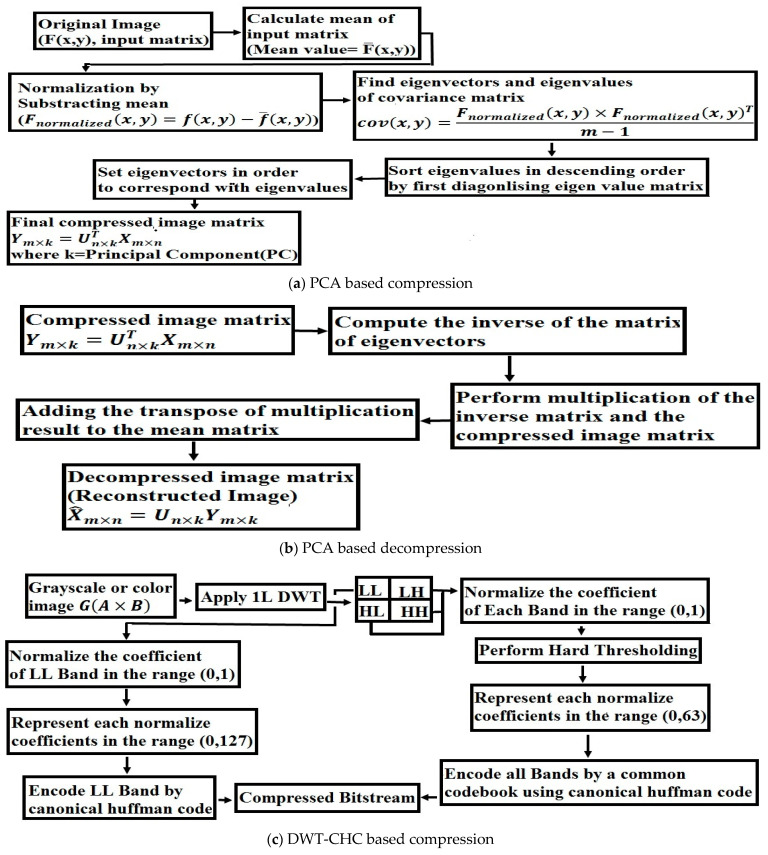
The proposed method is illustrated through flowcharts (**a**–**e**).

**Figure 5 entropy-25-01382-f005:**
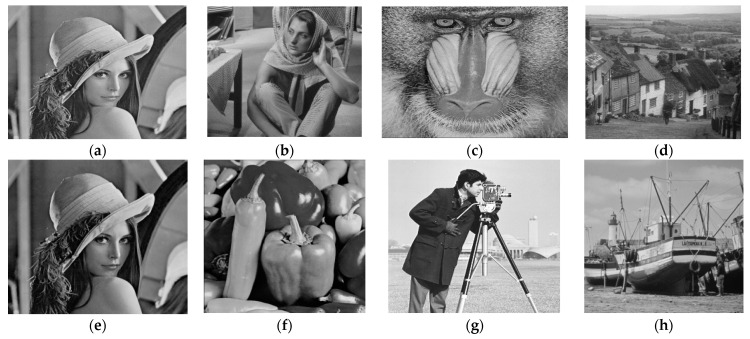
Test images in grayscale for size 512 × 512 (**a**–**d**) and 256 × 256 (**e**–**h**).

**Figure 6 entropy-25-01382-f006:**
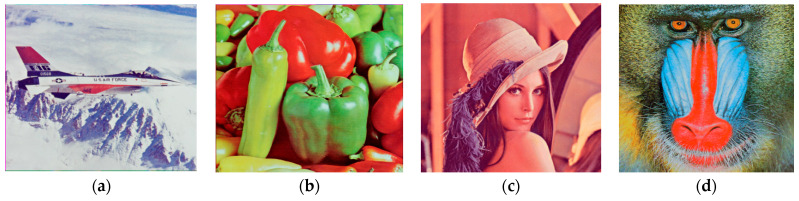
The size 512 × 512 (**a**–**e**) and 256 × 256 (**f**–**h**) (color test images).

**Figure 7 entropy-25-01382-f007:**
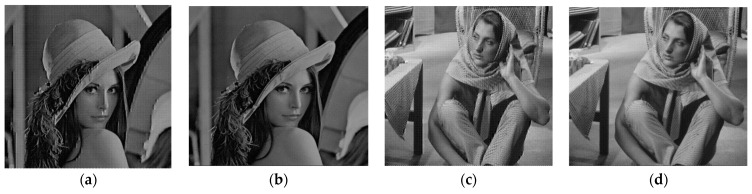
Results of compression for the 512 × 512 grayscale images of Lena and Barbara. (**a**) Lena image reconstruction using DWT with PSNR = 29.9001 dB and rate (bpp) = 3.2855; (**b**) Lena image reconstruction using the proposed method with PSNR = 34.7809 dB and rate (bpp) = 1.8158; (**c**) Barbara image reconstruction using DWT with PSNR = 27.7496 dB and rate (bpp) = 3.7896; (**d**) Barbara image reconstruction using the proposed method with PSNR = 33.3092 dB and rate (bpp) = 1.9806.

**Figure 8 entropy-25-01382-f008:**
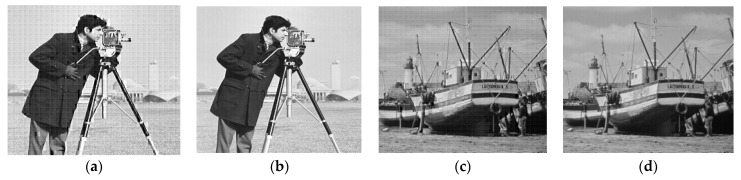
Compression outcomes for the grayscale images Cameraman and Boat of size 256 × 256: (**a**) reconstructed Cameraman image using DWT with PSNR = 26.4333 dB, rate (bpp) =2.7925; (**b**) reconstructed Cameraman image using proposed method with PSNR = 3 3.4238 dB, rate (bpp) = 1.5536; (**c**) reconstructed Boat image using DWT with PSNR=29.6486 dB, rate (bpp) = 3.4099; (**d**) reconstructed Boat image using proposed method with PSNR=37.9922 dB, rate (bpp) = 1.7985.

**Figure 9 entropy-25-01382-f009:**
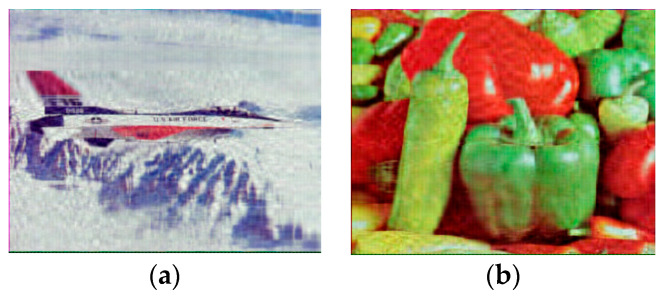
Results of compression for the 512 × 512 color images Airplane and Peppers. (**a**) Reconstructed image of an airplane using the proposed method, PSNR = 47.57 dB and rate (bpp) = 0.27; (**b**) Peppers image reconstruction using the proposed method, PSNR = 47.99 dB and rate (bpp) = 0.32.

**Figure 10 entropy-25-01382-f010:**
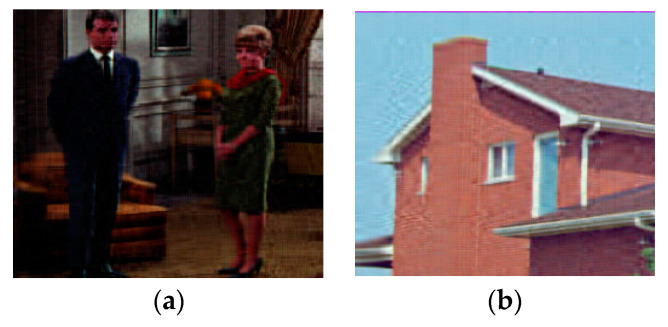
Results of color image compression for Couple and House with 256 × 256-sized images. (**a**) Reconstructed image of a couple with PSNR of 54.60 d B and rate (bpp) = 0.69; (**b**) reconstructed image of a house with PSNR of 53.47 dB and rate (bpp) = 0.70.

**Figure 11 entropy-25-01382-f011:**
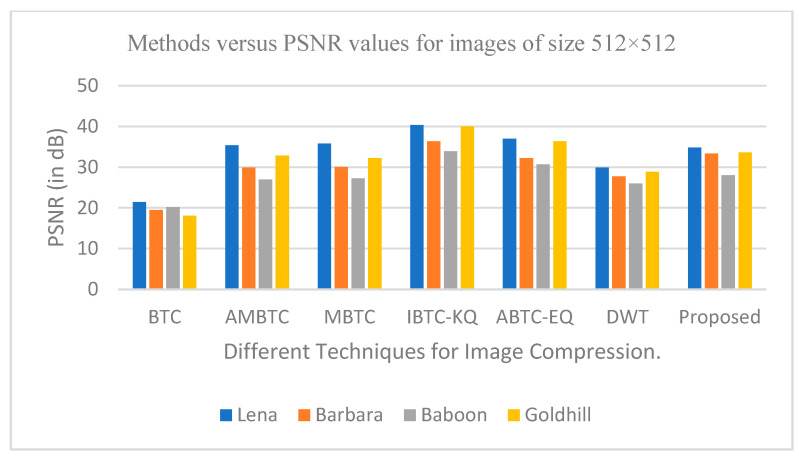
Comparison of various compression techniques used on the different test grayscale images (Lena, Barbara, Baboon and Goldhill).

**Figure 12 entropy-25-01382-f012:**
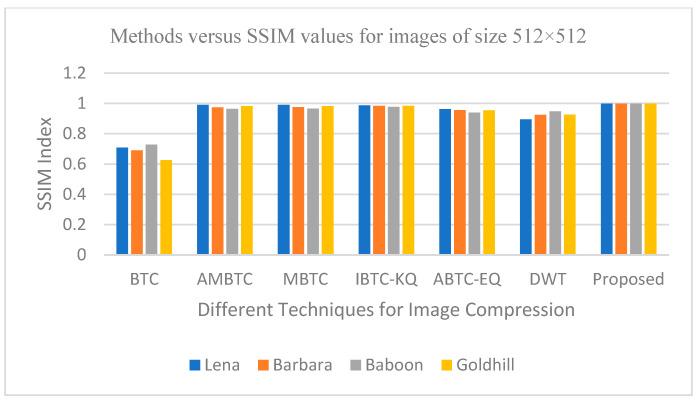
Comparison of various compression techniques used on the different test grayscale images (Lena, Barbara, Baboon and Goldhill).

**Figure 13 entropy-25-01382-f013:**
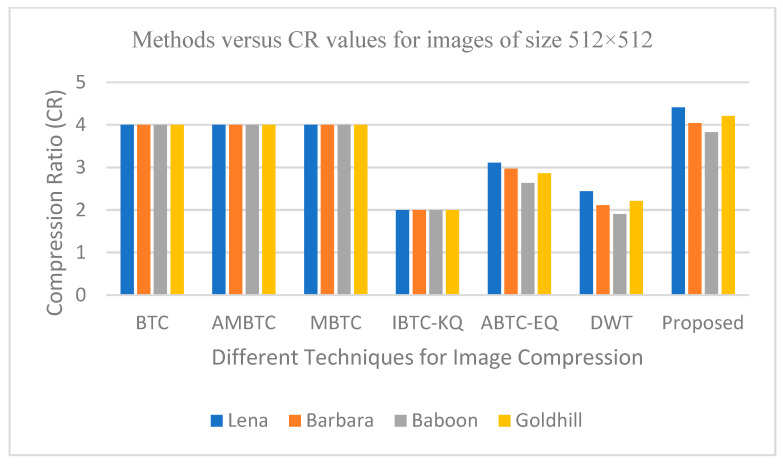
Comparison of various compression techniques used on the different test grayscale images (Lena, Barbara, Baboon and Goldhill).

**Figure 14 entropy-25-01382-f014:**
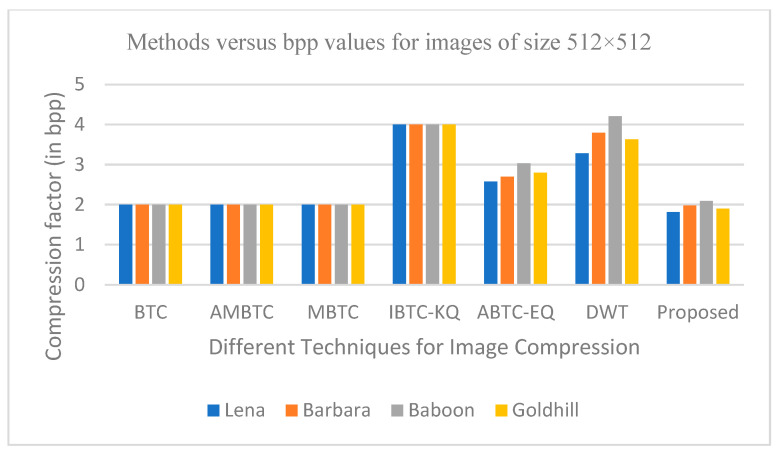
Comparison of various compression techniques used on the different test grayscale images (Lena, Barbara, Baboon and Goldhill).

**Figure 15 entropy-25-01382-f015:**
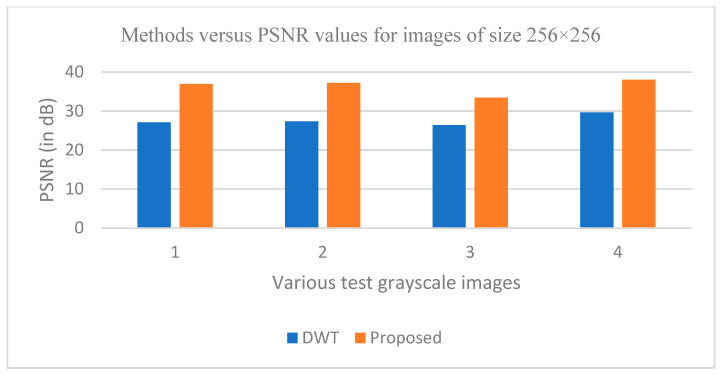
Comparison of various compression techniques used on the different test grayscale images: 1: Lena, 2: Peppers, 3: Cameraman and 4: Boat.

**Figure 16 entropy-25-01382-f016:**
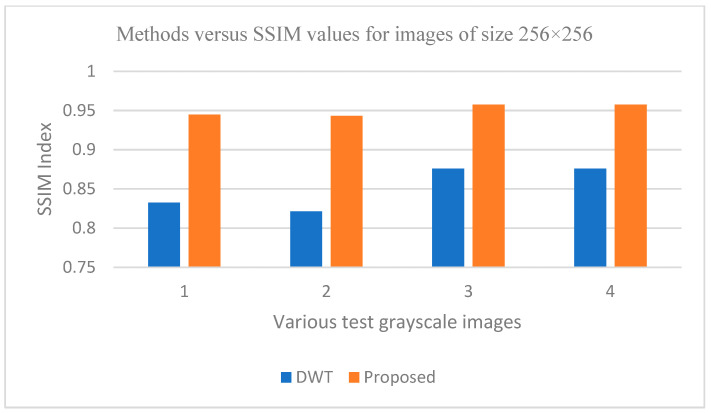
Comparison of various compression techniques used on the different test grayscale images: 1: Lena, 2: Peppers, 3: Cameraman and 4: Boat.

**Figure 17 entropy-25-01382-f017:**
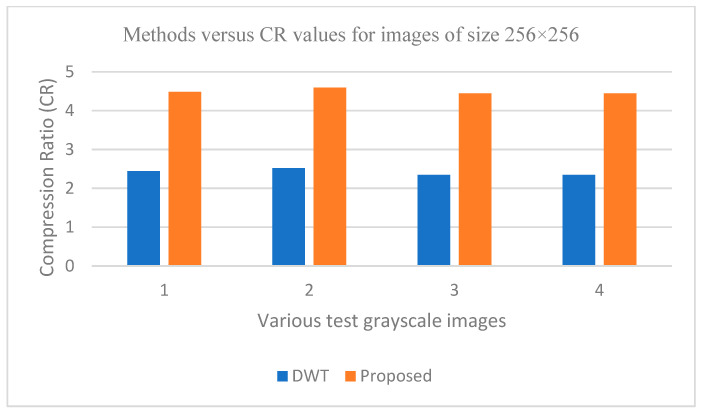
Comparison of various compression techniques used on the different test grayscale images: 1: Lena, 2: Peppers, 3: Cameraman and 4: Boat.

**Figure 18 entropy-25-01382-f018:**
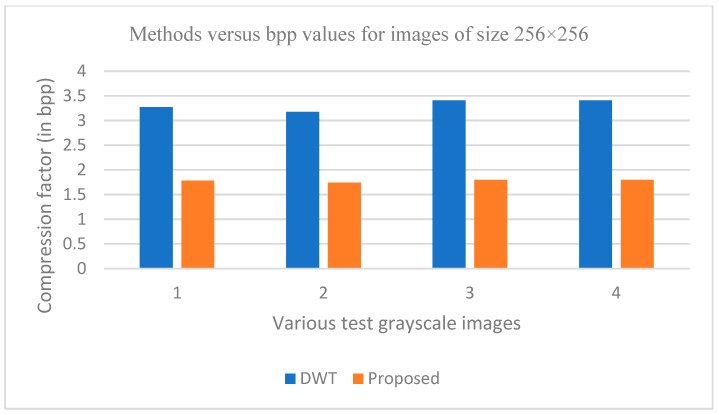
Comparison of various compression techniques used on the different test grayscale images: 1: Lena, 2: Peppers, 3: Cameraman and 4: Boat.

**Figure 19 entropy-25-01382-f019:**
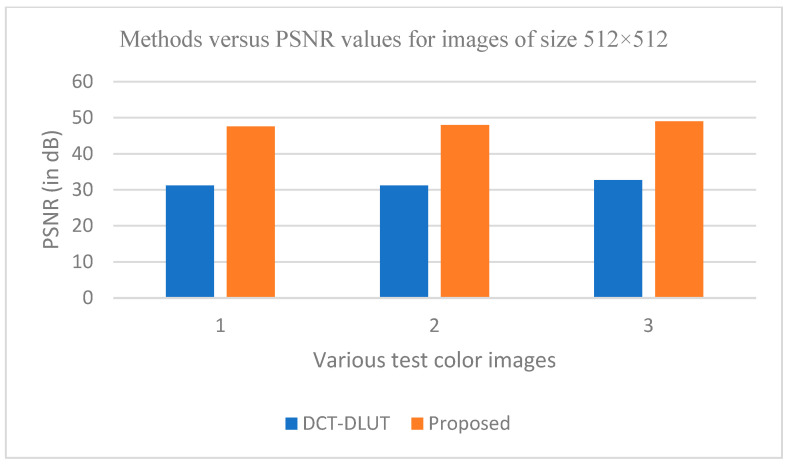
Comparison of various compression techniques used on the different color test images: 1: Airplane, 2: Pepper and 3: Lena.

**Figure 20 entropy-25-01382-f020:**
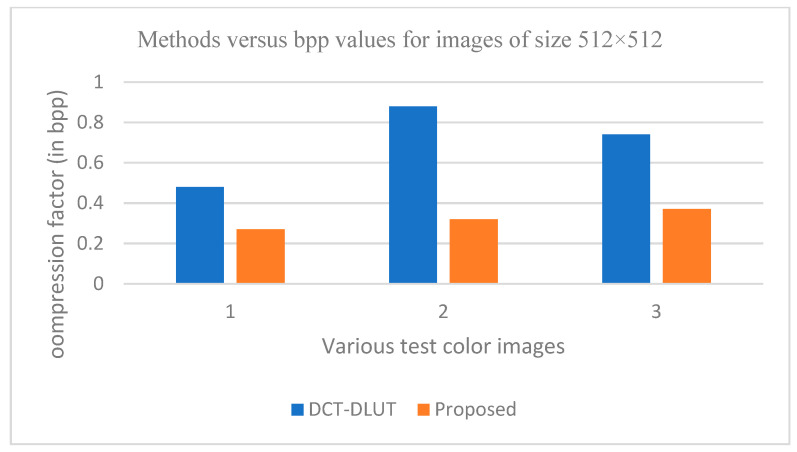
Comparison of various compression techniques used on the different color test images: 1: Airplane, 2: Pepper and 3: Lena.

**Figure 21 entropy-25-01382-f021:**
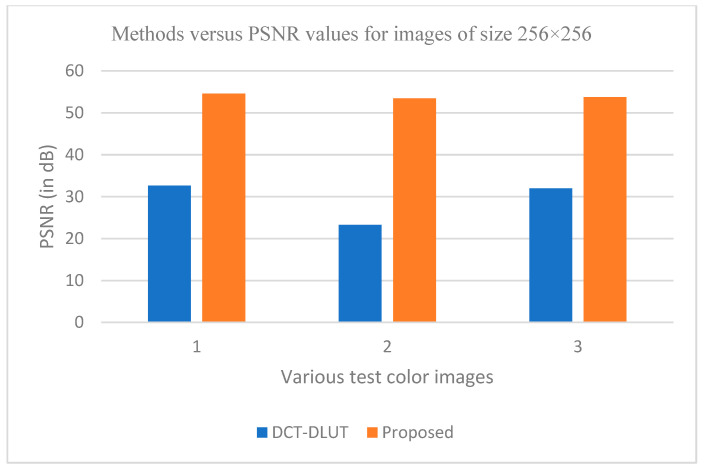
Comparison of various compression techniques used on the different color test images: 1: Couple, 2: House and 3: Zelda.

**Figure 22 entropy-25-01382-f022:**
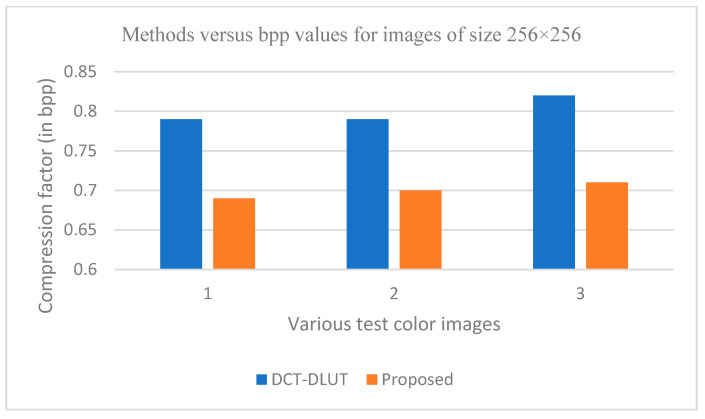
Comparison of various compression techniques used on the different color test images: 1: Couple, 2: House and 3: Zelda.

**Table 1 entropy-25-01382-t001:** Comparative performance of BTC [5], AMBTC [6], MBTC [7], IBTC-KQ [8], ABTC-EQ [9], DWT [16] and proposed method for grayscale images.

Tested Image	Method	Block Size (4 × 4) Pixels				Block Size (8 × 8) Pixels			
		PSNR	SSIM	BPP	CR	PSNR	SSIM	BPP	CR
Lena (512 × 512)	BTC	21.4520	0.7088	2	4	21.4520	0.7088	1.2500	6.4000
	AMBTC	35.3706	0.9905	2	4	32.0885	0.9639	1.2500	6.4000
	MBTC	35.8137	0.9904	2	4	32.6268	0.9662	1.2500	6.4000
	IBTC-KQ	40.3478	0.9874	4	2	36.4511	0.9664	2.5000	3.2000
	ABTC-EQ	36.9919	0.9632	2.5734	3.1087	33.8401	0.9305	1.8267	4.3794
	DWT	29.9001	0.8943	3.2855	2.4349	29.9001	0.8943	3.2855	2.4349
	Proposed	34.7809	**0.9985**	**1.8158**	**4.4058**	34.7809	**0.9985**	1.8158	4.4058
Lena (256 × 256)	DWT	27.0772	0.8326	3.2713	2.4455	27.0772	0.8326	3.2713	2.4455
	Proposed	**36.9556**	**0.9447**	**1.7831**	**4.4865**	**36.9556**	**0.9447**	**1.7831**	**4.4865**
Barbara (512 × 512)	BTC	19.4506	0.6894	2	4	19.4506	0.6894	1.2500	6.4000
	AMBTC	29.8672	0.9747	2	4	27.8428	0.9429	1.2500	6.4000
	MBTC	30.0710	0.9757	2	4	28.1069	0.9451	1.2500	6.4000
	IBTC-KQ	36.3729	0.9847	4	2	33.5212	0.9632	2.5000	3.2000
	ABTC-EQ	32.1986	0.9551	2.6966	2.9667	30.5587	0.9244	1.9487	4.1053
	DWT	27.7496	0.9242	3.7896	2.1111	27.7496	0.9242	3.7896	2.1111
	Proposed	33.3092	**0.9986**	**1.9806**	**4.0392**	33.3092	**0.9986**	1.9806	4.0392
Baboon (512 × 512)	BTC	20.1671	0.7288	2	4	20.1671	0.7288	1.2500	6.4000
	AMBTC	26.9827	0.9639	2	4	25.1842	0.9181	1.2500	6.4000
	MBTC	27.2264	0.9653	2	4	25.4677	0.9216	1.2500	6.4000
	IBTC-KQ	33.8605	0.9777	4	2	31.2925	0.9550	2.5000	3.2
	ABTC-EQ	30.6787	0.9400	3.0363	2.6348	28.7947	0.9089	2.1571	3.7086
	DWT	25.9806	0.9479	4.2012	1.9042	25.9806	0.9479	4.2012	1.9042
	Proposed	28.0266	**0.9984**	2.0917	3.8247	28.0266	**0.9984**	2.0917	3.8247
Goldhill (512 × 512)	BTC	18.0719	0.6252	2	4	18.0719	0.6252	1.2500	6.4000
	AMBTC	32.8608	0.9825	2	4	29.9257	0.9438	1.2500	6.4000
	MBTC	32.2422	0.9828	2	4	30.3195	0.9472	1.2500	6.4000
	IBTC-KQ	39.9867	0.9840	4	2	36.1776	0.9599	2.5000	3.2000
	ABTC-EQ	36.3085	0.9536	2.7986	2.8586	33.6061	0.9210	2.0778	3.8502
	DWT	28.8597	0.9255	3.6259	2.2064	28.8597	0.9255	3.6259	2.2064
	Proposed	33.6289	**0.9986**	**1.9020**	**4.2061**	**33.6289**	**0.9986**	1.9020	4.2061
Peppers (256 × 256)	BTC	19.4540	0.6306	2	4	19.4540	0.6306	1.2500	6.4000
	AMBTC	30.5655	0.9409	2	4	26.7127	0.8547	1.2500	6.4000
	MBTC	31.1372	0.9444	2	4	27.4445	0.8596	1.2500	6.4000
	IBTC-KQ	-----------	---------	--------	---------	-----------	---------	--------	---------
	ABTC-EQ	32.0306	0.9551	2.6966	2.9667	28.9805	0.8985	2.6966	4.0499
	DWT	27.3524	0.8212	3.1735	2.5209	27.3524	0.8212	3.1735	2.5209
	Proposed	**37.1723**	0.9431	**1.7422**	**4.5918**	37.1723	**0.9431**	1.7422	4.5918
Cameraman (256 × 256)	BTC	20.7083	0.7214	2	4	20.7083	0.7214	1.2500	6.4000
	AMBTC	28.2699	0.9322	2	4	25.8654	0.8831	1.2500	6.4000
	MBTC	29.0746	0.9392	2	4	26.9365	0.8934	1.2500	6.4000
	IBTC-KQ	36.7714	0.9890	4	2	33.6339	0.9754	2.5000	3.2
	ABTC-EQ	33.9790	0.9725	2.6418	3.0282	31.2452	0.9531	1.8325	4.3656
	DWT	26.4333	0.7483	2.7925	2.8648	26.4333	0.7483	2.7925	2.8648
	Proposed	33.4238	0.8578	**1.5536**	**5.1492**	33.4238	0.8578	1.5536	5.1492
Boat (256 × 256)	DWT	29.6486	0.8758	3.4099	2.3461	29.6486	0.8758	3.4099	2.3461
	Proposed	**37.9922**	**0.9575**	**1.7985**	**4.4482**	**37.9922**	**0.9575**	**1.7985**	**4.4482**

The bold letters represent the improved results among various reported works.

**Table 2 entropy-25-01382-t002:** Comparative performance of proposed method and DCT-DLUT [3] for color images.

Image	Proposed Method		DCT-DLUT	
	PSNR	BPP	PSNR	BPP
Airplane (512 × 512)	**47.57**	**0.27**	31.16	0.48
Peppers (512 × 512)	**47.99**	**0.32**	31.19	0.88
Lena (512 × 512)	**48.95**	**0.37**	32.65	0.74
Couple (256 × 256)	**54.60**	**0.69**	32.62	0.79
House (256 × 256)	**53.47**	**0.70**	23.27	0.79
Zelda (256 × 256)	**53.74**	**0.71**	32.01	0.82
Average	**59.71**	**0.51**	35.81	0.75

The bold letters represent the improved results among various reported works.

**Table 3 entropy-25-01382-t003:** Comparative performance of proposed method and NE-EZW [20] for color images.

Image	Proposed method		NE-EZW	
	PSNR	BPP	PSNR	BPP
Lena (512 × 512)	**48.95**	**0.37**	36.30	0.50
Peppers (512 × 512)	**47.99**	**0.32**	28.79	0.50
Mandrill (512 × 512)	**43.94**	**0.41**	34.20	0.50
House (512 × 512)	**46.80**	**0.31**	35.03	0.50
Average	**46.92**	**0.35**	33.58	0.50

The bold letters represent the improved results among various reported works.

**Table 4 entropy-25-01382-t004:** Time complexity of proposed method.

Image (256 × 256)	Running Time (s)	Running Time (s)
	Proposed	DWT [16]
Boat	92.3117	168.8943
Cameraman	87.1738	109.7213
Goldhill	110.1841	134.7754
Lena	96.6797	147.5008
Average	96.587325	140.22295

## Data Availability

Not applicable.

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
