# Peer review of "An Improved Image Compression Algorithm Using 2D DWT and PCA with Canonical Huffman Encoding"

_entropy, 2023, doi:10.3390/e25101382_

Round 1

Reviewer 1 Report

See the attached pdf file.

I do not like the phrases "the present authors" and "the present study" in Section 1. I hope that the authors replace them with "we" or "this paper".

Reviewer 2 Report

Review for the paper :

An Improved Image Compression Algorithm Using 2D DWT and PCA with Canonical Huffman Encoding

This papers suggests a new scheme for lossy image compression based on PCA, followed by DWT and CHC to enhance compression efficiency. The authors show better compression efficiency with compared to existing image compression algorithms both in term of objective image quality measure, such as PSNR and SSIM and image compression size ; CR and bbp.

I found some main problems with this manuscript < the main problem is the lossy part which is the  PCA transformation. It is well known that PCA is the best transform from compression both in term of energy compaction and decorrelation, however it was not adapted by the standards committee such as JPEG due to fact of complexity (with compared to other transformation such as DCT) , and mostly due to the Inverse PCA transform at the decoder side. In order to utilize the inverse PCA transform the decoder needs to know the eigen vectors of the original image and if you choose to transmit it to the decoder side then you need to waste a lot of bits for transmission this information. Therefore the PCA is usually not used image or video compression, moreover it was found that DCT is mostly behave very similar to PCA while the basis vector are cosine functions that are well known to the decoder ( no need to transmit them) .

The authors didn’t indicate in the decompression side the inverse PCA transform at all – and this is a missing peace in this work – an elaboration on that particular part is missing and very important for the whole success of their algorithm.

Also in the comparison section there is no comparison to the well known and most popular image compression standard JPEG and also JPEG 2000.

The authors also present their results in term of quality and bitrate separately. But usually when you compare different coding algorithm you must present your results in term of Rate Distortion curves which are not presented in the manuscript. Also in all figures when comparing quality in term of SSIM and PSNR you must indicate the rate (bbp) for the comparison, and for graphs of CR you must indicate the quality for all images.

So for conclusion , in order to consider this manuscript for publication , the main indicated problems that I mentioned above should be considered and major revision is needed before considering publication at Entropy journal.

 Moderate editing of English language required

Round 2

Reviewer 1 Report

I admit that the authors have earnestly improved the presentation of their article very well.  All of the mandatory conditions from me have been satisfied.

None.

Reviewer 2 Report

The manuscript looks better now but still i have two comments :

1. How does the decoder perform the inverse PCA transform on the image ? Eigenvectors of the the encoded image should be transmitted along with the compressed image - and this is not considered in the manuscript (additional bandwidth ).

2. The Rare Distortions curves don't make sense  , please checks the graphs ( Figure 11 to Figure 15) - these graphs are not legitimate. Please correct.
